# The Lung Microbiome during Health and Disease

**DOI:** 10.3390/ijms221910872

**Published:** 2021-10-08

**Authors:** Kazuma Yagi, Gary B. Huffnagle, Nicholas W. Lukacs, Nobuhiro Asai

**Affiliations:** 1Department of Pathology, University of Michigan Medical School, Ann Arbor, MI 48109, USA; nlukcas@med.umich.edu; 2Department of Pulmonary and Critical Care Medicine, Department of Internal Medicine, University of Michigan Medical School, Ann Arbor, MI 48109, USA; ghuff@umich.edu; 3Department of Microbiology and Immunology, University of Michigan Medical School, Ann Arbor, MI 48109, USA; 4Mary H. Weiser Food Allergy Center, University of Michigan Medical School, Ann Arbor, MI 48109, USA

**Keywords:** lung microbiome, host-microbe interactions, asthma, chronic obstructive pulmonary disease (COPD), bronchiectasis, lung cancer, respiratory viral infections

## Abstract

Healthy human lungs have traditionally been considered to be a sterile organ. However, culture-independent molecular techniques have reported that large numbers of microbes coexist in the lung and airways. The lungs harbor diverse microbial composition that are undetected by previous approaches. Many studies have found significant differences in microbial composition between during health and respiratory disease. The lung microbiome is likely to not only influence susceptibility or causes of diseases but be affected by disease activities or responses to treatment. Although lung microbiome research has some limitations from study design to reporting, it can add further dimensionality to host-microbe interactions. Moreover, there is a possibility that extending understanding to the lung microbiome with new multiple omics approaches would be useful for developing both diagnostic and prognostic biomarkers for respiratory diseases in clinical settings.

## 1. Introduction

Healthy human lungs have been traditionally considered to be sterile or free from bacteria for a long time [1,2]. However, since lungs have constant contact with the environment, they are continually exposed to microbes. Moreover, lower airway are warm, moist surfaces centimeters away from the oral and nasal cavities that are bacteria-rich environments [1], with microaspiration of pharyngeal contents to be well-documented to occur in healthy or asymptomatic subjects using both culture-dependent and -independent techniques [1,3,4,5]. Culture-independent molecular techniques for microbial identification such as pyrosequencing have also demonstrated that large numbers of microbial organisms, including bacteria, fungi and viruses, collectively known as the lung microbiome, exist in the lung of both healthy subjects and those with respiratory diseases [2,5].

The lungs and airways harbor diverse composition of microbes that are not only detected but undetected by conventional culture-based techniques. To date, many studies have confirmed that microbial communities in respiratory disease are different from healthy subjects, and the lung microbiome can not only influence susceptibility or causes of respiratory diseases but be affected by disease activities of respiratory diseases as well as in responses to treatment.

In this review, we will summarize the impact of the lung microbiome during health as well as both acute and chronic respiratory diseases such as asthma, chronic obstructive pulmonary disease (COPD), bronchiectasis, lung cancer, and respiratory viral infections.

## 2. The Respiratory Tract Microbiome during Health

### 2.1. The Lungs Are Not Sterile and Are Not the Gut

Although the dogma that the healthy human lung is free from bacteria has persisted until very recently [1,2], many studies have demonstrated that microaspiration is common in healthy, asymptomatic patients [3,4,6]. In 2001, Lederberg et al. suggested that the importance of “the ecological community of commensal, symbiotic, and pathogenic microbes that literally share our body space and have been ignored as determinants of health and disease” [7]. Numerous studies using culture-independent molecular techniques have demonstrated that a diverse bacterial community is present in the lower airway of healthy condition, and have identified the main genera as *Prevotella*, *Veillonella*, and *Streptococcus* [2,5,8,9,10,11,12,13,14,15]. The lung microbiome in healthy subjects shows a rich, diverse bacterial community that are present in low abundance [16]. Many functions of microbes are important for human health, as reported by previous studies using gnotobiotic mice models, including, protection from invading pathogens [17], modulation of immune system [18], and nutrient uptake [19].

In healthy lungs, the microbial biomass is much lower (10^3^ to 10^5^ bacteria per gram of tissue [14]) than that in the lower gastrointestinal (GI) tract (10^11^ to 10^12^ bacteria per gram of tissue [20]). Although the GI tract and lungs are both mucosa-lined luminal organs sharing embryological origin, micro-anatomical features are quite different. Microbes in the GI tract is unidirectional from mouth to the anus in the absence of vomiting or esophageal reflux. Microbes introduced orally must survive both the acidic pH of the stomach and the alkaline pH of the duodenum to immigrate into the cecum. On the other hand, the movement of air, mucus, and microbes in the lungs are not unidirectional but bidirectional [1,21,22]. The microbiome of the lungs is more dynamic and transient than that of the GI tract. The GI tract is of uniform temperature (37 °C) throughout its entire length, while the epithelial surfaces of the respiratory tract have a gradient from ambient temperature to core body temperature in the alveoli [1,23]. Moreover, the lungs are oxygen rich (aerobic) during health unlike the GI tract (anerobic). Similar to the GI tract, while the epithelial surfaces of the trachea and bronchi are covered with mucus, the majority of the alveoli is covered with a lipid-rich surfactant that has bacteriostatic against select bacterial species [24]. Furthermore, the GI tract and lungs have different host-bacterial interactions. While luminal IgA levels are much higher in the GI tract [25], lungs have more extraluminal interactions between bacteria and host alveolar macrophages [26]. Therefore, these differences of environmental conditions between the GI tract and the lungs results in divergent microbial communities.

### 2.2. Key Ecological Factors Determining the Lung Microbiome

It is considered that the composition of the lung microbiome is determined by the balance of three factors: [27] (1) microbial immigration into the airways, (2) elimination of microbes from the airways, and (3) the relative reproduction rates of its community members found in the airways, which is determined by the regional growth conditions (Figure 1). Microbial immigration is driven by inhalation of microbes from air, subclinical microaspiration of upper respiratory tract and oral cavity, and direct dispersion along airway mucosal surface [3,4,6,28]. Microbial elimination is a function of mucociliary clearance, cough, and host (both innate and adaptive) immune defense systems. The ecological factors that shape the regional growth conditions of the lung microbiome include pH, temperature, oxygen tension, and nutrient availability as well as local microbial competition, host epithelial cell interactions, and activation of host inflammatory cells. While the lung microbiome is determined by the balance of immigration and elimination in health condition [15,29,30], the regional growth conditions of the lung microbiome change and alter dramatically during disease. These alterations result in disease- and patient-specific microbial communities optimized for the injured airways [22,27].

### 2.3. The Oral Microbiome and the Nasal Microbiome

Since microaspiration of pharyngeal contents is well-documented to occur in healthy or asymptomatic subjects [1,3,4,5], many culture-independent studies have reported that the microbial communities of the lungs more closely resemble those of the oropharynx than inhaled air, the nasopharynx [12,13,30]. Several studies have demonstrated that nasal microbiome has a minimal effect on lung communities in healthy conditions, whereas nasal microbiome is closer to skin microbiome [9,30]. Other studies have indicated sampling lungs with introduced bronchoscope intranasally had minimal influence of upper respiratory tract contamination on acquired specimens [27,31]. Taken together, it is considered that the oral microbiome is the primary source of the lung microbiome during health.

## 3. The Lung Microbiome during Disease

The composition of the lung microbiome is determined by the balance of microbial immigration, elimination, and regional growth conditions as mentioned above. Both acute and chronic respiratory disease can dramatically change these three key ecological factors [21,27]. Many studies have compared the lung microbiome during disease with those of healthy subjects, and have found significant differences in composition in patients comparing health and disease situations [32,33,34,35], and it has been also shown that disease is associated with a loss of bacterial diversity, or by the dominance of a single taxon or small group of taxa [36]. In this section, we will highlight findings of association between the lung microbiome and respiratory diseases such as asthma, COPD, bronchiectasis, lung cancer, and respiratory viral infections.

### 3.1. Asthma

Asthma is the most common chronic and heterogeneous respiratory disease, afflicting over 300 million adults and children; 250,000 die each year [37,38,39]. Recent studies have expanded our understanding that bacterial composition in the airway is associated with disease severity in asthmatics, and dysbiosis of the lung microbiome can play an important role in the pathogenesis of asthma. Analysis of lung microbiome from bronchial brushings samples displayed that asthmatics who receive corticosteroid therapy are dominated by Proteobacteria phylum, with a mix of potential pathogens, including *Haemophilus*, *Moraxella* and *Neisseria* [40]. Response to steroids is accompanied by an increase in *Actinobacteria*, whereas *Klebsiella* is linked to severe asthma [40]. As for the relation between airway microbial composition and asthma phenotype, Taylor and Simpson et al., using sputum samples documented that patients with neutrophilic asthma, who receive high doses of inhaled corticosteroids (ICSs), demonstrate a less diverse bacterial load with relative enrichment in *Haemophilus* and *Moraxella* species (sp.) and a reduction in the relative abundance of *Streptococcus*, *Gemella*, and *Porphyromonas* taxa compared with patients with eosinophilic asthma [41,42]. It may also be important to consider what types of samples tested can influence the results of microbiome composition. Induced sputum and bronchoalveolar lavage fluid (BALF) samples are quite different, as Durack et al. previously reported [43]. In addition, it has been shown that the lung-gut axis also correlated with the pathogenesis of asthma. Epidemiologic studies have revealed several potentially protective environmental factors, such as growing up on a farm, vaginal birth, breast-feeding, the presence of household pets, birth order, and the number of children, as well an increased risk of asthma being associated with antibiotic use during late pregnancy and the first year of life. These factors are strongly associated with gut dysbiosis. It has been shown that specific bacterial taxa can contribute to reduce the risk of asthma in a prospective longitudinal cohort study among Canadian babies. *Lachnospira*, *Veillonella*, *Faecalibacterium* and *Rothia* in the gut of 3-month-old infants correlated with elevated risk of developing asthma [44]. Moreover, a case–control study in Ecuador displayed that gut dysbiosis in early infancy is associated with childhood atopic wheeze (AW). Fecal samples taken from children with AW contained a higher proportion of *Streptoccoccu* sp., *Bacteroides* sp., and *Pichia kudriavzevii*, and a lower proportion of *Ruminococcus* sp. and *Bifidbacterium* sp. [45]. A pilot study demonstrated a significant relationship between the gut microbiome and lung function in a group of well-characterized asthmatic and non-asthmatic adults [44]. We might consider gut dysbiosis to be a helpful optional tool for the treatment of asthma.

### 3.2. Chronic Obstrutive Pulmonary Disease (COPD)

COPD is a life-threatening lung disease that causes an economic and health burden worldwide. Like other respiratory diseases, increasing evidence suggests that the lung microbiome can play an important role in acute exacerbation of COPD [46,47,48]. Studies comparing respiratory microbiome in BALF and sputum between COPD patients and healthy subjects have identified a change of microbial diversity with an increased relative abundance of *Moraxella*, *Streptococcus*, *Proteobacteria*, *Veillonella*, *Eubacterium*, and *Prevotella* sp. in disease [49,50]. Conversely, Sze et al. performed a study comparing GOLD stage 4 COPD patients and a control group using explanted lung tissue and reported an increase in Proteobacteria phylum organisms and a decrease in Firmicutes, Bacteroidetes phylum, *Streptococcus*, *Haemophilus infuenzae*, and *Prevotella* sp. Have also been documented [51]. Some of the above studies suggested that these alternations might be independent of smoking history and that microbial diversity might be more apparent in examining association in specific COPD endotypes [50]. During an acute exacerbation event, fecal microbiome has also been shown to display a lower relative abundance of Firmicutes and Actinobacteria, with an associated increase in Bacteroidetes and Proteobacteria [52,53]. These results suggest that the gut-lung axis can affect the disease severity of COPD, and maintenance of lung and gut microbiome could be a useful preventive option for acute exacerbation of COPD.

### 3.3. Bronchiectasis

Bronchiectasis is characterized radiologically by permanent enlargement of the airways, and clinically by cough, sputum production and recurrent respiratory infections [54]. Bronchiectasis is a heterogenous disease and is associated with multiple underlying etiologies [54,55,56]. Pathophysiological mechanisms contain persistent bacterial infections, dysregulated immune responses, impaired mucociliary clearance, and airway obstruction [54,55,57]. Although bronchiectasis is previously classified as a rare or orphan disease, the diagnosis of bronchiectasis is increasing worldwide [58,59]. To date, no approved treatment exists for the condition other than for bronchiectasis caused by cystic fibrosis (“non-CF bronchiectasis”) [55,56], patients with non-CF bronchiectasis need to be categorized according to a heterogeneous group by endotype or by clinical phenotype to better describe patients [57]. Chronic bacterial colonization and infection is a characteristic feature in patients with non-CF bronchiectasis [54,55,57]. Since the presence of some organisms such as *Pseudomonas aeruginosa* (*P. aeruginosa*) is associated with prognosis or frequency of exacerbation [60,61], sputum culture has had an important role of management. Traditional culture methods have shown that *P. aeruginosa* and *Haemophilus influenzae* (*H. influenzae*) are isolated most frequently, but other Gram-negative (*Moraxella catarrhalis*, *Escherichia* sp., and *Klebsiella* sp.) and Gram-positive bacteria (*Streptococcus pneumoniae* and *Staphylococcus aureus*) are also isolated [62]. With the advent of sequencing technologies that allow more comprehensive profiling of the bacterial composition in the lungs, the understanding of chronic infection in patients with non-CF bronchiectasis is evolving. As with other respiratory diseases, several studies in patients with non-CF bronchiectasis has revealed that loss of bacterial diversity, or by the dominance of a single taxon or small group of taxa is associated with its disease activity. It has been reported that the Shannon-Wiener diversity index (measure of richness and evenness, or composite diversity) shows a positive linear correlation with lung function in patients with non-CF bronchiectasis [63]. According to the results using traditional culture methods, studies using sequencing technologies have confirmed that the proteobacteria, which include *Pseudonomas* and *Haemophilus* genera, come to dominate the diseased airway in patients with non-CF bronchiectasis and have been associated with neutrophil-mediated inflammation and frequency of exacerbations [36,63,64,65]. Rogers et al. have reported that *P. aeruginosa*-dominated and *H. influenzae*-dominated communities had significantly worse lung function and higher levels of interleukin (IL)-8 and IL-1β [64]. Moreover, several studies have also reported that a subgroup of non-CF bronchiectasis patients with microbiome dominated with firmicutes such as anaerobe *Veillonella* genera have frequent exacerbations in spite of a lower degree of neutrophilic inflammation [66,67,68]. Pathologic changes of the composition in the lung microbiome could occur as a result of antibiotic exposure in patients with non-CF bronchiectasis. A long-term study of macrolide (erythromycin) treatment (the BLESS study) revealed a decreased overall diversity of the microbiome and increase in the relative abundance of *Pseudomonas aeruginosa* as a consequence of the reduced relative abundance of other microbes sensitive to macrolides [68].

Although bacteria have received the most attention in studies of the lung microbiome in non-CF bronchiectasis, fungi and mycobacteria also contribute to its disease activity and progression. *Aspergillus* is the taxa that most greatly differs between healthy subjects and patients with non-CF bronchiectasis, and the abundance of *Aspergillus* has been associated with exacerbations. These findings suggest that *Aspergillus* could also have a key role to airway inflammation in non-CF bronchiectasis [69]. Pulmonary infection due to nontuberculous mycobacteria (NTM) is one of the causes of non-CF bronchiectasis [54,55,56], and is an emerging public health concern worldwide [70,71]. Several studies have suggested that the disease progression of NTM and antimicrobial therapy might influence the stable state of the lung microbiome including mycobacteriome and contribute to further dysbiosis and clinical progression [35,72,73,74], while it is still controversial whether this finding is a cause of the disease progression or consequence of disease or the treatment. Although these studies mentioned above have used 16S ribosomal RNA (rRNA) analysis for detecting NTM, this approach has inherent limitations including underestimation of mycobacteria [73].

As with other respiratory diseases, to date, bacteria, viruses, fungi have been considered as separate entities, while the true lung microbiome or multi-ome consists of all microbes and their genes, including bacteria, viruses and fungi. Emerging approaches such as metagenomics allow comprehensive understandings of bacterial, viral, and fungal communities simultaneously [75,76].

### 3.4. Lung Cancer

A significant amount of recent evidence has emerged and found a correlation between the gut microbiome and cancer [77,78,79,80]. There has been relatively little evidence that the pathogenesis of lung cancer is linked to the lung microbiome. However, recent reports displayed that the clinicopathogenesis of lung cancer can be correlated with the lung microbiome. The most common form of lung cancer is non-small cell lung cancer (NSCLC), which is subdivided into three main types as follows: adenocarcinoma, squamous cell carcinoma (SCC), and large cell undifferentiated carcinoma [81,82]. Of these, lung cancer patients with adenocarcinoma show a different microbiome from those with SCC. According to the taxonomic analyses, the lung microbiome among lung cancer patients, at the phylum level, was particularly enriched in Proteobacteria, Firmicutes, Acinobacteria, Bacteroides, and Verrucomicrobia. Of these, Firmicutes was the most abundant. *Prevotella* was dominant on lung microbiota components at the genus level, together with *Bifidobacterium*, *Acinetobacter* and *Ruminococcus* by BALF samples [78,79]. Huang et al. documented the correlation between lung microbiome and histology, as well as risk of disease progression. *Streptococcus* was significantly lower in metastatic adenocarcinoma than that in non-metastatic adenocarcinoma, and *Veillonella* and *Rothia* were higher in metastatic SCC compared to non-metastatic SCC using bronchial washing fluid samples [80]. Another study showed that *Rothia* was more abundant in SCLC than NSCLC [79,80].

Lung cancer remains a leading cause of cancer death, and approximately 13% of all patients with lung cancer are diagnosed with small cell lung cancer (SCLC). Despite high sensitivity to first-line chemotherapy, the prognosis is still disappointing for clinicians [83]. A few studies assessed the correlation between lung microbiome and clinicopathogenesis of SCLC. Both saliva and BALF samples were enriched in *Treponema*/*Spirochetes* and *Pseudomonas*, respectively [84]. Compared to NSCLC, *Rothia* was more abundant in SCLC by BALF samples [85].

Of note, recent evidence has demonstrates that gut dysbiosis might impair the systemic response to efficacy in advanced cancer patients. Studies have documented that it is possible to predict based upon their gut microbiome whether the cancer patients are hyper-responder or non-responder to immune checkpoint inhibitor-based immunotherapy [86]. The alpha diversity between the gut and respiratory microbiota was not linked, and only the gut microbiota alpha diversity was associated with anti-programmed cell death-1 response. Higher gut microbiota alpha diversity indicated better response and more prolonged progression-free survival [86]. Antitumor immunotherapy is promising, however, very costly, leading to an increase in the social-economic burden. In addition, since examining feces is not invasive testing, analysis of the gut microbiome could be used to aid in the selection of antitumor treatment for cancer patients.

### 3.5. Respiratory Viral Infections

Lung and gut dysbiosis are correlated with disease severity and progression of respiratory viral infections. The emergence of a novel coronavirus, severe acute respiratory syndrome coronavirus 2 (SARS-CoV-2), and the pandemic of coronavirus disease 2019 (COVID-19) is ongoing worldwide [87,88]. It is well-known that the disease severity and mortality rate of COVID-19 are different by ages comorbidities. Many kinds of comorbidities have been known to be associated with lung and gut dysbiosis, such as asthma, COPD, cancer, diabetes mellitus, and metabolic disorders. Therefore, it is hypothesized that dysbiosis can predict the disease severity of COVID-19 [89,90].

Gaibani et al. performed a study comparing the lung microbiome between COVID-19 patients who are critically ill, and COVID-19 negative pneumonia patients by BALF samples. COVID-19 patients had a lower microbial diversity with a significantly higher relative abundance of *Pseudomonas* sp. compared to COVID-19-negative subjects. On the contrary, the lung microbiome in COVID-19-negative pneumonia patients was mainly characterized by the enrichment of *Haemophilus influenzae*, *Veillonella dispar*, *Granulicatella* sp., *Porphyromonas* sp., and *Streptococcus* sp. [91]. Several studies have already reported that respiratory virus infections can alter the microbial composition and total amount of bacterial load [21,92]. Commensal bacteria such as *Prevotella* sp., *Veillonella* sp. characterize the microbial composition in healthy persons and are involved in the maintenance of the host immune homeostasis. Thus, it is reasonable that viral infection induced-alternation of microbial composition and the total amount of bacterial load can contribute to disease severity in patients with respiratory viral infections.

Recent studies showed that gut microbiome composition differed significantly among COVID-19 patients compared with non-COVID-19 individuals [91,92]. Besides, plasma concentrations of inflammatory cytokines, chemokines, and tissue damage markers are correlated with gut microbiota composition. Yeoh et al. also assessed which specific species enriched or depleted in COVID-19 patients were correlated to inflammatory markers concentration. The results showed that *Bifidobacterium adolescentis*, *Eubacterium rectale*, and *Faecalibacterium prausnitzii* play immunomodulatory roles in the human GI system, which are negatively correlated with concentrations of inflammatory makers among the patients. Conversely, *Bacteroides dorei* and *Akkermansia muciniphila* were positively correlated with IL-1β, IL-6, and C-X-C motif ligand 8 (CXCL8) [93].

As for other respiratory viruses, the correlation between lung and gut microbiome and influenza virus has been explored. Gu et al. conducted a cross-sectional study to compare the gut microbiomes by feces samples among patients with influenza virus (H1N1) infection, COVID-19, and healthy control (HC). The result displayed a significant higher relative abundance of opportunistic pathogens, such as *Streptococcus*, *Rothia*, *Veillonella*, and *Actinomyces*, and a lower relative abundance of beneficial symbionts [94]. A significant decrease in gut microbial diversity and abundance in H1N1 patients as well as COVID-19 patients, compared to HCs. The gut microbiome in the H1N1 patients was enriched by *Enterococcus*, *Prevotella*, *Finegoldia*, and *Peptoniphilus*, whereas the microbiome of HCs dominated *Blautia*, *Romboutsia*, *Collinsella*, *Bifidobacterium*, and other beneficial bacteria. In addition, compared with the gut microbiome between H1N1 and COVID-19 patients, the relative abundance of *Prevotella*, *Ezakiella*, *Murdochiella*, and *Porphyromonas* was higher in the H1N1 group than in COVID-19 patients. This latter study supported the concept that the alternation of gut microbiome differs by different viral infections [94]. Influenza virus infection causes severe respiratory inflammation, resulting in 500,000 annual deaths worldwide [95] despite widespread vaccines and effective anti-viral drugs. Some clinical trials using probiotics to modulate the dysbiosis of the lung and gut are ongoing [96,97]. Modulation of the lung and gut microbiome could enhance anti-viral responses and reduce the detrimental pathophysiology that occurs during and after respiratory viral infections and aid in more effective therapy.

## 4. Perspectives of Lung Microbiome Research

### 4.1. Limitations of Lung Microbiome Research

To date, almost all the lung microbiome studies previously reported are observational with a study design using cross-sectional comparisons with clinical variables. Although research on the lung microbiome has confirmed that microbial composition in respiratory disease such as asthma, COPD, bronchiectasis, lung cancer, and respiratory viral infections are different from healthy subjects as mentioned in this review, descriptions of the lung microbiome alone provide insufficient insight into mechanisms. Moreover, there are no strict, prescriptive guidelines for microbiome study design, experimental execution, detection methods (16S rRNA gene sequencing, shotgun metagenomics gene sequencing, and whole-genome sequencing), analysis of microbiome data, and reporting [16,98,99]. Therefore, each study design should be considered with acceptable and well-established principles of experimental methodology, feasibility, and sampling. We need to clarify the limitations we have faced such as utilization of well-described specimens (e.g., sputum, specimens from bronchoscopy, lung tissue from surgical excision), deep characterization of the specific host response to microbial composition both locally and systemically, and experimental models in which host -microbe interactions are modulated to better understand potential mechanisms.

### 4.2. Host-Microbe Interactions

As mentioned in this review, the microbiome can induce a host response, and lead to enhanced disease. One of strategies to clarify host-microbe interactions is to show the correlation between major microbiome features such as relative operational taxonomic unit (OTU) abundance and host immune response using cellular or molecular disease markers [51]. Segal et al. suggested a pneumotype that had higher rRNA gene concentration and higher relative abundance of supraglottic-characteristic taxa, such as *Veillonella* and *Prevotella*, was associated with enhanced subclinical lung inflammation [100], and that pneumotype was also associated with a distinct metabolic profile, enhanced expression of inflammatory cytokines, a pro-inflammatory phenotype characterized by elevated T-helper cell type 17 (Th 17) lymphocytes. [13]. Wu et al. confirmed that a single episode of aspiration of nonpathogenic bacteria (*Streptococcus mitis*, *Veilonella parvula*, and *Prevotella melaninogenica*) led to temporal changes in lower-airway dysbiosis, a prolonged increase in lung inflammatory tone, and a decrease in the susceptibility to subsequent *Streptococcus pneumoniae* infection [101]. They also revealed that the resulting alteration of lung inflammatory tone lasted at least 14 days and evoked sustained host immune responses including Th1 and Th17 activation, inflammasome, P38 mitogen-activated protein kinase (MAPK), and phosphoinositide 3-kinase (PI3K)/AKT signaling pathways [101]. Although it is still unclear whether the protection against secondary pneumococcal infection is pathogen specific or not, these results have provided how the most common microbial exposure in lung affects host immune responses and susceptibility to other respiratory pathogens.

To further extend our understanding, multiple omics approaches are needed [98]. A study using paired lung sample from patients with lung cancer run simultaneously for 16S rRNA sequencing and transcriptome profiling have revealed genes (extracellular signal-regulated kinase [ERK]) and pathway (PI3K pathway) associated with microbial composition [102]. Moreover, to integrate the microbiome with DNA methylation, proteomics and metabolomics can add further dimensionality to host-microbe interactions [103,104]. These multiple omics approaches will provide functional information, which is absent from 16S rRNA gene sequencing.

### 4.3. Clinical Applicatiton of the Lung Microbiome

According to the finding from previous studies and this review, the lung microbiome might contain both diagnostic and prognostic information, and it is expected that the lung microbiome would become one of the useful biomarkers for respiratory diseases in clinical settings. Our ultimate goal of lung microbiome research is to discover key diagnostic or therapeutic features that impact clinical outcomes, and to achieve precision medicine. While the gut microbiome might be modifiable via diet to alter the gut-lung axis, it is still unclear whether the lung microbiome itself can be of therapeutic benefit in respiratory diseases. Although randomized, controlled trials have not confirmed a reduction in asthma incidence [105], there is a possibility that probiotics restoring a healthy gut microbiome might reduce Th2 cytokine responses in patients with allergic asthma [106]. Proper characterization of the lung microbiome of specific disease endotypes, clarification of endotype-related gene targets modulated by the lung microbiome, and development of novel methods to affect the lung microbiome are required to achieve a personalized therapeutic approach based on the lung microbiome, disease phenotype and the co-morbidities associated with the respiratory disease.

## Figures and Tables

**Figure 1 ijms-22-10872-f001:**
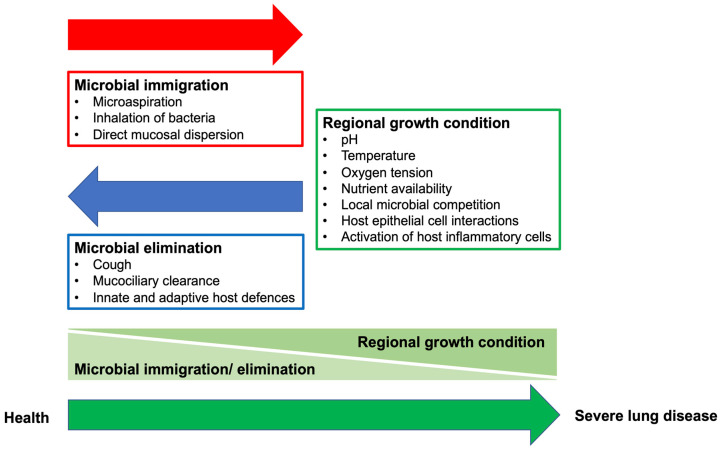
Key Ecological Factors Determining the Lung Microbiome: microbial immigration, microbial elimination, and the relative reproduction rates of its community members. The lung microbiome is determined mainly by microbial immigration and elimination in healthy subjects. In severe lung disease, the composition of lung microbiome is primarily determined by regional growth conditions. Adapted from Dickson et al., 2014 [27].

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
