# Peer review of "The Lung Microbiome during Health and Disease"

_ijms, 2021, doi:10.3390/ijms221910872_

Round 1

Reviewer 1 Report

The authors review the role of the lung microbial microbiome in health and disease. Most lung microbiome communities were not known previously, and significant differences have been found for the microbial composition in lungs of healthy persons and those suffering from respiratory disease. The lung microbiome plays a role in disease outcome and is also affected by systemic disease activities or treatments. Besides a direct role in respiratory disease and lung cancer, authors highlight evidence for a connection between lung infection and dysbiosis of the gut microbiome. This may be unexpected, however the phenomenon has recently been emphasised by a series of studies of patients with respiratory COVID and influenza, as listed in this review. The infection or dysbiosis of the lung microbiome increased (local) inflammation, as shown by activation of ERK and PI3K pathways, which in turn affected gut microbiome homeostasis. This effect may also work the other way and it is suggested that reducing dysbiosis of the gut microbiome, and thereby the inflammatory response, lung diseases such as asthma may be mitigated.

The authors conclude that improved standardisation of study design and a further understanding of the lung microbiome composition should lead to new diagnostic and prognostic biomarkers for respiratory diseases.

GENRAL COMMENTS

Several typos found and sometimes sentences miss woods (for examples see below). Please check carefully.

SPECIFIC COMMENTS
Pg 2 ln 84 Suggest: ‘…of microbes from the airways, and (3) the relative reproduction rates of its community members found in the airways,…’

Pg4 ln 155-7: Suggest ‘Fecal samples taken from children with AW contained a higher proportion of Streptoccoccus sp., Bacteroides sp., and Pichia kudriavzevii, and a lower proportion of Ruminococcus sp. and Bifidobacterium sp. [45]’

Pg 6 ln 256: Word missing in ‘Streptococcus was significantly lower in metastatic adenocarcinoma versus not (….??), and Veillonella and Rothia…’

Pg 7 ln 295-7 Suggest: ‘Commensal bacteria such as Prevotella sp. and Veillonella sp. characterize the microbial composition in healthy persons and are involved in the maintenance of the host immune homeostasis. ‘

Streptococcus and T whipplei should be considered as pathogens.

Pg 7 ln 309 Suggest: ‘Conversely, Bacteroides dorei and Akkermansia muciniphila were positively correlated with…’

Pg 8 ln 338 Suggest: ‘Moreover, there are no strict, prescriptive guidelines for microbiome study design, experimental execution,…’

Pg 8 ln 377: The sentence ‘As the highlighted in these studies, it is important to design translational research with paying attention to plan and resource allocation.’ does not make sense. Please rephrase.

Author Response

Dear reviewer 1,

Thank you very much for your great and prompt reviewing. We totally agree with your opinions and made appropriate revisions according to your suggestions. The revised portion was red in the text. We strongly believe that this revision made the text better than before. Please, consider my article for publication in your journal.

Sincerely,

Nobuhiro Asai, Sep 24, 2021

Reviewer 2 Report

In this study the authors summarize the impact of the lung microbiome during health as well as both acute and chronic respiratory diseases such as asthma, chronic obstructive pulmonary disease (COPD), bronchiectasis, lung cancer, and respiratory viral infections and the host-microbe interaction. The review is well written and defined, the references are also abundant and sufficiently updated. I appreciated the separation of the article into 4 paragraphs, especially the last paragraph where the authors talk about future prospects and the host microbiome interaction.

The microbiome is certainly a topic of great interest that deserves further study and multiple approaches.

I have no specific comments to do.

Author Response

Thank you so much for your reviewing. I really appreciate your great help.